# Developing a Psychological Research Methodology for Evaluating AI-Powered Plush Robots in Education and Rehabilitation

**DOI:** 10.3390/bs15101310

**Published:** 2025-09-25

**Authors:** Anete Hofmane, Inese Tīģere, Airisa Šteinberga, Dina Bethere, Santa Meļķe, Undīne Gavriļenko, Aleksandrs Okss, Aleksejs Kataševs, Aleksandrs Vališevskis

**Affiliations:** 1Institute of Digital Humanities, Riga Technical University, 12/1 Azenes Street, LV-1048 Riga, Latvia; airisa.steinberga@rtu.lv; 2Center for Pedagogy and Social Work, Riga Technical University Liepaja Academy, 14 Liela Street, LV-3401 Liepaja, Latvia; dina.bethere@rtu.lv (D.B.); santa.melke@rtu.lv (S.M.); undine.gavrilenko@rtu.lv (U.G.); 3Institute of Architecture and Design, Riga Technical University, 6 Kipsalas Street, LV-1048 Riga, Latvia; aleksandrs.okss@rtu.lv (A.O.); aleksandrs.valisevskis@rtu.lv (A.V.); 4Institute of Mechanical and Biomedical Engineering, Riga Technical University, 6 Kipsalas Street, LV-1048 Riga, Latvia; aleksejs.katasevs@rtu.lv

**Keywords:** autism spectrum disorders, AI-powered plush robots, social rehabilitation

## Abstract

The integration of AI-powered plush robots in educational and therapeutic settings for children with Autism Spectrum Disorders (ASD) necessitates a robust interdisciplinary methodology to evaluate usability, psychological impact, and therapeutic efficacy. This study proposes and applies a four-phase research framework designed to guide the development and assessment of AI-powered plush robots for social rehabilitation and education. Phase 1 involved semi-structured interviews with 13 ASD specialists to explore robot applications. Phase 2 tested initial usability with typically developing children (N = 10–15) through structured sessions. Phase 3 involved structured interaction sessions with children diagnosed with ASD (N = 6–8) to observe the robot’s potential for rehabilitation, observed by specialists and recorded on video. Finally, Phase 4 synthesized data via multidisciplinary triangulation. Results highlighted the importance of iterative, stakeholder-informed design, with experts emphasizing visual properties (color, texture), psychosocial aspects, and adjustable functions. The study identified key technical and psychological evaluation criteria, including engagement, emotional safety, and developmental alignment with ASD intervention models. Findings underscore the value of qualitative methodologies and phased testing in developing child-centered robotic tools. The research establishes a robust methodological framework and provides preliminary evidence for the potential of AI-powered plush robots to support personalized, ethically grounded interventions for children with ASD, though their therapeutic efficacy requires further longitudinal validation. This methodology bridges engineering innovation with psychological rigor, offering a template for future assistive technology research by prioritizing a rigorous, stakeholder-centered design process.

## 1. Introduction

The deployment of AI-powered plush robots in educational and rehabilitation settings, especially for children with Autism Spectrum Disorders (ASD), requires a robust interdisciplinary foundation. These robots have been designed and evaluated to enhance social, emotional, and cognitive development in children with ASD through personalized and interactive experiences ([10]; [22]). Studies emphasize the importance of integrating psychological, clinical, and engineering approaches to ensure usability, ethical acceptability, and therapeutic efficacy ([7]; [11]). Additionally, systematic reviews have documented the effectiveness of robot-assisted therapy in improving behaviors such as joint attention and imitation ([2]). Furthermore, research with the humanoid robot Kaspar has demonstrated its efficacy in therapy for children with ASD ([51]), and the rapid deployment of mobile applications during the COVID-19 pandemic highlighted the critical role of technology in maintaining therapeutic accessibility ([13]), strengthening the contextual relevance of this study. The authors of the present publication have developed innovative AI-powered plush robots designed for effective use within social rehabilitation and educational settings for children with ASD. These include four types of toys: a cartoon-like cat, a realistic-looking cat, a cartoon-like dog, and a realistic-looking dog. Provisionally, four kits will be manufactured and distributed to educational centers for children with ASD that are involved in the research. With recent technological advances, it has become possible to design sophisticated and reliable artificial robots to support therapeutic interventions for children with ASD.

For autism therapy to be successful, such robots must be capable of interacting with children in a personalized way and adapting to their individual needs through the application of AI techniques, as children with ASD often demonstrate behaviors that differ from those of their typically developing (TD) peers. Smart sensory toys also help to address this challenge, as they are already familiar to children and therefore require no special adaptation or programming.

The purpose of this publication is to present a research methodology, grounded in psychological approaches, for studying AI-driven plush robots in the context of children with ASD, with a particular focus on rehabilitation and educational environments. To achieve this objective, the article seeks to address two research questions:How should research be designed to study AI-powered plush robots amongst children with ASD?What technical and psychological concepts should be assessed whilst evaluating usability for children with ASD in rehabilitation and educational environments?

This study is theoretically grounded in theoretical frameworks for human–technology interaction, as well as in current research about plush robot introduction to children. Regarding theories about human–technology interactions, three complementary frameworks were analyzed: the Mobile App Rating Scale (MARS) ([47]), the DIPEx criteria for patient-centered technologies, and the ISO 9241-210 standard ([26]) on Human-Centered Design.

### 1.1. MARS: Quality and Engagement Framework

Originally developed for assessing mobile health applications, the MARS framework provides structured criteria, engagement, functionality, aesthetics, and information quality that are directly applicable to evaluating the psychological usability of interactive robots ([47]). For instance, engagement measures how effectively the robot captures a child’s attention—an essential factor in child–robot interaction research ([28]; [24]). Likewise, functionality assesses responsiveness to user input such as touch or voice, aligning with frameworks for socially assistive robot design ([12]). Studies adapting MARS-like criteria to health apps report that functionality and aesthetics often score higher than engagement or information quality, illustrating where design improvements may focus ([20]). These domains map directly onto psychological theories of motivation and usability, supporting their use as a robust, interdisciplinary foundation for evaluating AI-powered plush robots in pediatric contexts.

### 1.2. DIPEx: Ethical and Experiential Evaluation

The DIPEx (Database of Individual Patient Experiences) methodology is grounded in rigorous qualitative narrative interviewing and maximum-variation sampling, offering a transparent, inclusive approach to collecting lived experiences across diverse populations ([25]). Originating in the UK and disseminated internationally, DIPEx synthesizes patient or caregiver stories through lay-language thematic summaries, illustrated with multimedia clips and participant-approved transcripts ([9]). This patient-centered framework aligns with key values of credibility, inclusivity, transparency, and authenticity, all of which are critical when designing empathetic and psychologically safety-focused interactive technologies. In robot-mediated systems, such as AI-powered plush robots used in pediatric rehabilitation, DIPEx-inspired principles ensure emotional safety, respect for neurodiversity, and lived-experience relevance, reinforcing a design approach that centers actual user needs, particularly those of children with ASD and their caregivers ([45]).

### 1.3. ISO 9241-210: Human-Centered Design for Psychological Usability

The ISO 9241-210 standard emphasizes core principles of human-centered design—including iterative evaluation, multidisciplinary collaboration, active user involvement, and attention to the whole user experience ([26]). Applying these principles in robot design underscores the need to consider not only technical usability but also sensory, emotional, and cognitive factors, such as tactile softness, facial expressiveness, and verbal communication, that influence a child’s psychological engagement and comfort. The standard further supports integrating psychological evaluation techniques, such as observational analysis and user interviews, throughout design and testing phases to ensure emotional safety and user-centered relevance ([26]; [32]).

### 1.4. Alignment with ASD Intervention Models

The evaluation approach is also informed by widely recognized ASD therapy frameworks such as TEACCH (Treatment and Education of Autistic and Related Communication Handicapped Children), ABA (Applied Behavior Analysis), and SCERTS (Social Communication, Emotional Regulation, and Transactional Support). These models emphasize structured support, predictability, and sensory modulation, features that can be translated into the robot’s behavior and content design. Incorporating these intervention principles ensures the robot not only serves as a playful tool but also supports therapeutic and educational goals aligned with psychological and developmental needs ([36]; [38]).

### 1.5. Psychological Usability in Human–Robot Interaction

Beyond these frameworks, the theoretical foundation is further enriched by the field of human–robot interaction (HRI), which examines how humans perceive, emotionally relate to, and behaviorally engage with robotic systems ([14]). Plush robots are an emerging class of therapeutic technology in pediatric rehabilitation. As these plush robotic devices are introduced to help children with physical or cognitive challenges, researchers rigorously evaluate two key aspects: usability and psychological impact. In scientific studies, a multi-method evaluation framework is common, combining quantitative scales with qualitative observations to capture a holistic picture. Observational methods are widely used; for instance, [39] ([39]) analyzed video recordings to evaluate interaction levels and robot task adherence. Similarly, [15] ([15]) employed semi-structured interviews with parent–child dyads to explore emotional responses and engagement levels. Structured tools like the System Usability Scale ([4]) or the Mobile App Rating Scale ([46]) are commonly used in quantitative evaluations. Studies also use customized engagement scores ([29]) or observational coding of social behaviors ([21]) to evaluate child–robot interaction quality. Emotional and cognitive outcomes are often measured with validated psychological instruments, such as the Wong–Baker FACES scale for pain ([50]) or the Revised Children’s Anxiety and Depression Scale ([8]). These studies collectively inform a robust theoretical framework for assessing plush robots in child rehabilitation, ensuring interventions are psychologically meaningful and user centered.

## 2. Materials and Methods

Prior to data gathering, this research received ethical approval from the Riga Technical University ethics committee. The research consisted of 4 phases. The following section will address each of the phases, describing procedure, sample and instruments.

### 2.1. Preliminary Data Gathering Semi-Structured Expert Interviews

The task of this stage was to explore the representations professionals have regarding the usability, psychological characteristics, and potential applications of AI-powered plush robots for children with ASD. The sample consisted of 13 experts with at least two years of experience in social rehabilitation and education of children with ASD, holding higher education degrees, with prior use of technology in their work, and willingness to participate (inclusion criteria). Data was collected through semi-structured interviews with specialists to understand the potential applications of AI-powered sensory plush robots for children with ASD. The interview protocol used in this study is detailed in its entirety in Appendix A, and was designed to elicit nuanced perspectives across five broad domains. These domains must be perceived as guiding sections for information gathering rather than as predefined analytic categories: (1) specialists’ experience context (e.g., prior experience with technology, qualifications), (2) perceptions regarding the robots’ use in rehabilitation (e.g., perceived advantages, ideal use cases), (3) socio-psychological aspects of child–robot interaction (e.g., initial reactions, social skill development), (4) visual parameters of the robots (e.g., geometric dimensions, suitability of fur length), and (5) robot functions (e.g., necessity of eye reactions, presence reminders). Interviewers were instructed to employ non-leading language and encouraged to probe for detailed responses. Data collection commenced following informed consent, and recordings were subsequently transcribed verbatim. After collecting all interviews, data were analyzed using reflexive thematic analysis ([3]). To ensure analytic rigor and minimize subjectivity, and to enhance trustworthiness through data triangulation ([34]), we employed a code reconciliation process. Three researchers independently coded data, then collaboratively reviewed categories and subcategories. All initial codes were examined. Any shared codes with each examiner, that met a consensus were taken. If discrepancies arose, they were addressed through structured discussion, a ‘code reconciliation session’, guided by the principles of (1) verifying that all researchers were interpreting the same data segments, addressing potential terminological differences that masked underlying agreement, (2) presenting and defending individual interpretations with supporting quotes from the interview transcripts, with researchers potentially deferring to one expert’s interpretation based on compelling evidence, or (3) pursuing a ‘merged interpretation,’ wherein the researchers collaboratively developed a revised code or theme that encapsulated the perspectives of all analysts. This rigorous and transparent process strengthened the credibility and validity of our findings, ensuring a robust foundation for interpreting the expert perspectives. No intercoder reliability coefficients (such as Cohen’s kappa or Krippendorff’s alpha were calculated, as reflexive thematic analysis does not assume the use of a fixed coding frame. Rigor was supported through triangulation and collaborative theme refinement ([3]).

### 2.2. Pilot Testing with Typically Developing Children

The second phase aimed to test the initial usability and interaction flow of the AI- The second phase aimed to test the initial usability and interaction flow of the AI-powered plush robots with children in the target age range without ASD. After the semi-structured expert interviews were presented to the research team, adjustments to the AI-powered plush robots were made, and some sensor clusters were identified. ‘Sensor clusters’ refer to grouped tactile sensors on the robot’s surface (e.g., on the paw, back) programmed to trigger specific auditory or visual responses. These sensory clusters were then coded and piloted amongst 10 to 15 typically developing children aged between 4 and 7 years in educational settings. The goal was to assess the initial usability and interaction patterns of the robot outside the ASD population before formal testing. The inclusion criteria required that the child be within the specified age range and have no known developmental diagnoses. Informed consent was obtained from parents or legal guardians, and the children were introduced to the robot in a supervised, structured setting. Observations, which were analyzed to define interaction clusters, were captured via digital observation protocol. ‘Interaction clusters’ refer to predefined sequences of child–robot interactions (e.g., ‘initiate touch,’ ‘respond to sound’) used to structure the sessions and for observational coding. This observational data was then used to program and define these interaction clusters for subsequent phases.

### 2.3. Interaction Sessions with Children with ASD

The aim of the third phase was to evaluate the educational and rehabilitation potential of the AI-powered plush robots in structured sessions with children diagnosed with ASD. AI-powered plush robots were introduced to a sample of 6 to 8 children formally diagnosed with ASD. Participants were recruited through collaborations with specialized rehabilitation centers and preschools implementing special education programs across three Latvian cities (institution names withheld for confidentiality), identified through established professional networks. Inclusion required confirmed ASD diagnoses, verified either by a child psychiatrist using the Autism Diagnostic Observation Schedule (ADOS) for rehabilitation center participants or a Pedagogical Medical Commission assessment for preschool attendees. Exclusion criteria minimized confounding variables, specifically excluding children who had not yet completed their initial adaptation period at their institution as determined by specialist assessment, due to potential stress. A purposive sampling strategy ensured representation across functional levels and verbal abilities in a stable, representative sample. Given that these children were recruited from rehabilitation centers in collaboration with specialists already working with them, a sense of trust was already established. Each session, lasting approximately 10–15 min and conducted in a familiar, quiet therapy room to minimize distractions, involved specialists who had undergone a 1 h training session covering the robot’s functions, the study’s goals, ethical considerations, and how to facilitate interactions without prompting specific behaviors. Session fidelity was monitored post-session by a research team member reviewing the video recordings. To supplement the qualitative data gathered during the interaction sessions, we also administered the System Usability Scale ([4]) to the specialists immediately following each session. Children participated in 2 to 4 structured sessions with the robot, with the sessions being recorded on video for later analysis. Specialists then completed structured observation protocols post-session. Detailed observation protocols are accessible to be viewed in Appendix B. This allowed for direct coding with the previous information presented.

### 2.4. Psychological Profiling

The final phase of the study focused on synthesizing the data collected from all previous stages. This involved a triangulated analysis of video recordings, observation protocols, interview transcripts, and anecdotal feedback from specialists and parents. At first all the observational protocols were analyzed separately, with each source examined for recurring patterns relevant to the study aims and the child. A multidisciplinary team of three researchers with backgrounds in education, psychology, and technology conducted independent coding of the qualitative data. Each expert generated thematic categories and subcategories, which were then discussed collaboratively to reach consensus. Following this, a triangulation approach was applied: the results from the three sources were compared to identify points of convergence (consistent patterns across data types) and divergence (unique observations in one source). Discrepancies were reviewed and interpreted with reference to established theoretical frameworks. The agreed-upon themes informed the development of psychological interaction profiles for each child and were used to adjust the behavioral and sensory programming of the AI-powered plush robots. This phase emphasized scientific rigor through methodological triangulation, expert collaboration, and iterative refinement of robot interaction design. Video recordings were essential for capturing the dynamics of children’s interaction with the AI-powered plush robots. While observation protocols documented the context and sequence of behaviors and dynamics, video data allowed to examine in detail how children touched, pressed, or manipulated the robots and how these actions activated responses in the toys. This enabled us to link particular forms of interaction (e.g., squeezing, stroking, tapping) with corresponding sensor responses. The information from video recordings and protocols provides behavioral and emotional immediate response to the AI-powered plush robots, feedback from parent’s and specialists provided information on social rehabilitation outcomes and dynamics such as communication. The integration of multiple data sources enabled the recognition of consistent behavioral and emotional response patterns, which were synthesized into individualized psychological profiles of the children’s interactions with the AI-powered plush robots.

## 3. Results

To answer the first research question: “How should research be designed in order to study AI-powered plush robots amongst children with ASD?” The results indicate that the interdisciplinary approach from social sciences and engineering is a successful way to study AI-powered plush robots in rehabilitation and educational settings amongst children with ASD (See Figure 1).

From the social sciences a qualitative research methodology seems to be the best approach in the research of AI-powered plush robots. It is important to note that data should be gathered from all 4 of the phases before making final adjustments to the plush robots (see Figure 2). However, each phase gave data that was valuable during the next of the phases.

The preliminary data yielded themes relevant to engineering discipline. It is important to note that topics yielded by thematic analysis at this stage uncover valuable information but must not be taken as definite and final opinion when it comes to making changes in the properties and functioning of the AI-powered plush robots. If all or majority of experts agree, then the changes can be noted, and if justified, then implemented. In most of the cases, experts had varying opinions, which was very important to note and reflect to the research team. It is important to sample experts from various backgrounds (education, psychology, speech therapy, physiotherapy, etc.) to give a diverse perspective, especially on what goals AI-powered plush robots can achieve for children with ASD in rehabilitation and education settings. Results also show how important it is to do everything step-by-step, starting with hypotheses from experts, then gathering data from children without ASD and only then, after educating the specialists about the AI-plush robots, their potential goals in educational and rehabilitation settings, design the interaction sessions. To ensure data validity it is important to gather the data with various methods: observation protocols, video recordings, semi-structures interviews.

Another result indicates that data should be gathered not just from the interaction between the child and specialist regarding the AI-powered plush robot, but also from primary caretakers, the specialist that has worked with the child and objective measures. The data from the children–specialist interaction is best acquired through video material and observation protocol the specialist fills after each session. While the protocol should be structured, and includes the same themes from the thematic analysis, it is important to leave room for notes about the session in which the specialist can describe in a free form. This allows for data triangulation to understand what should be improved in the robots. The data should also be gathered from the parents at the end of all the sessions, to understand what has changed for the child with ASD. Also interviews regarding overall feedback should be gathered from the specialist about each child at the end of all the sessions.

To answer the second research question: “What technical and psychological concepts should be assessed whilst evaluating usability for children with ASD in rehabilitation and educational environments?” the initial results were yielded from the first phase preliminary data, which carried through the observational protocol, feedback from primary caregivers and specialists (See Table 1).

This framework can be used throughout the data gathering keeping in mind that the specialist or primary care giver must provide the perspective from the child with ASD, their reaction, behavior, perception, etc. This can serve as a framework, keeping in mind that with different data gathering methods comes different emphases on constructs, e.g., in observation protocols, it is going to be more structured and will include ready-made answers more about the nature of interaction itself, rather than hypothetical observations. After all the data are gathered, psychological profiling can take place combining these categories.

## 4. Discussion

The present research proposes and highlights the value of qualitative methodology rooted in social sciences as a necessary first step for exploring the usability and potential impact of AI-powered plush robots in educational and rehabilitation contexts for children with ASD. This aligns with broader research indicating that qualitative approaches are well-suited to capture the nuanced, context-dependent experiences of users and stakeholders in emerging technological interventions ([17]; [3]).

Importantly, findings from all four phases of the research contributed unique insights, supporting the necessity of a step-by-step, iterative design process (see Figure 2). Each phase served not only as a standalone data point but also as a guidepost for the next, emphasizing the cyclical nature of design-based research in educational technology ([18]). The data gathered during preliminary expert interviews offered valuable input on the psychological, social, visual, and functional dimensions of the AI-powered plush robots. While these early insights are indispensable, it is essential to recognize that they should not be interpreted as definitive directives for robot design. Rather, they should serve as provisional guidelines, subject to reevaluation as more diverse data becomes available.

This staged research structure reflects principles of human-centered design, where stakeholder input, particularly from multidisciplinary experts such as educators, psychologists, speech therapists, and physiotherapists, is continually integrated ([33]; [44]). Divergent opinions among experts were especially revealing. Rather than being a limitation, such variation enhances the robustness of the research by highlighting context-specific concerns and goals. These finding echoes prior studies in the assistive technology domain which stress the importance of multidisciplinary perspectives in technology co-design for neurodiverse populations ([19]; [41]).

The results further underscore the importance of beginning with expert-generated hypotheses, followed by gradual, controlled data collection from interactions with children, first without ASD and only subsequently including children with ASD. This staged implementation safeguards ethical standards while allowing professionals to develop an informed understanding of the technology’s purpose and capabilities. Such sequential adaptation is supported by the literature on responsible innovation and adaptive trial design in educational technologies ([16]; [31]).

The need for methodological triangulation also emerged strongly. To ensure ecological validity and contextual richness, data should be gathered through various formats, including structured observation protocols, video analysis, and semi-structured interviews. This triangulation aligns with established practices in qualitative inquiry for strengthening validity and reliability ([34]). Moreover, allowing professionals to add free-form reflections within structured observation protocols was found to be critical in identifying subtle, emergent phenomena not captured by predefined categories—a practice consistent with recommendations in qualitative fieldwork ([48]).

Furthermore, the findings advocate for a comprehensive data collection strategy that extends beyond the child-specialist dyad. Collecting supplementary feedback from primary caregivers and professionals familiar with the child’s developmental history ensures a holistic understanding of how the AI-powered plush robot may influence not only targeted rehabilitation outcomes but also general well-being and behavioral patterns. Such multi-informant approaches are increasingly encouraged in autism research to reflect the full ecological system of the child ([30]; [35]).

It is crucial to emphasize that the primary contribution of this study is the validation of a qualitative, segmented methodology. The results indicate that these robots have a significant potential to support crucial therapeutic objectives; however, these observations are preliminary and are based on data collected during the design phase. They should be regarded as promising hypotheses that require further empirical validation.

The findings also align with other research, suggesting the potential of animal-like plush robots in supporting children with ASD, particularly through their social interaction capabilities and visually appealing properties. Research has indicated that social interaction could be enhanced when children engage with robots that respond to touch, gaze, and simple dialog ([5]; [40]; [27]). These interactions are hypothesized to promote shared attention, turn-taking, and emotional expression, which are key developmental goals in ASD therapy ([23]; [49]). Importantly, the animal-like form could support familiarity and emotional safety, which are believed to be crucial for building trust and reducing anxiety ([43]; [37]).

The visual design of these robots, including soft textures, rounded features, large eyes, and ambiguous zoomorphic forms, helps avoid overstimulation and promotes gentle engagement ([42]; [52]). Children are more likely to interact meaningfully with robots that are visually soft and emotionally neutral, especially when the robot avoids uncanny or overly mechanical aesthetics ([6]; [19]). This effect is further enhanced when robots offer responsive feedback through movement, sound, and facial expressions that align with a child’s actions ([53]; [1]).

Altogether, the synergy between visual properties and interaction mechanisms plays a critical role in shaping how children engage with AI-powered plush robots. The literature supports that child–robot interaction becomes most effective when both visual cues and social affordances are considered during design ([27]; [43]; [49]), lending preliminary support for further investigation into the role of such robots in rehabilitation and educational contexts.

In sum, this study provides a methodological framework for the future design and evaluation of AI-powered plush robots for children with ASD that emphasizes flexibility, evidence, and ethical considerations. The qualitative, phase-based approach employed here allows for gradual refinement and contextual sensitivity. The crucial next step is to transition from this methodological and design-focused phase to rigorous efficacy testing. Future studies should continue to emphasize stakeholder diversity, data triangulation, and longitudinal observation to empirically evaluate the therapeutic effectiveness of the robots developed through this framework. ensuring that these technologies are not only functional but also meaningful, safe, and developmentally appropriate.

## 5. Conclusions

In summary, this study demonstrates that a qualitative, interdisciplinary methodology is not only well-suited but essential for the rigorous evaluation of AI-powered plush robots for children with ASD. The proposed four-phase research design provides a robust and, crucially, ethical framework for assessing initial usability and therapeutic potential. By sequentially gathering data from experts (Phase 1), then typically developing children (Phase 2), and finally children with ASD in a specialist-guided setting (Phase 3), the methodology ensures safety, builds stakeholder trust, and allows for iterative refinement before full-scale deployment. The synthesis of this data through multidisciplinary triangulation (Phase 4) guarantees that findings are deeply rooted in real-world psychological and educational contexts.

The significant therapeutic potential of AI-powered plush robots for ASD intervention, through enhanced engagement, predictability, and customizability, necessitates an equally robust, psychologically grounded evaluation methodology. Without such a framework, there is a tangible risk of deploying technologies that are technologically sophisticated yet therapeutically ineffective or counterproductive. Our proposed four-phase framework is an essential step toward ensuring these tools are developed and implemented in a manner that is truly beneficial, ethical, and centered on the complex needs of children with ASD.

The practical application of this methodology offers a clear pathway for developers, clinicians, and educators. For developers, it provides a template for child-centered design, highlighting key evaluation criteria such as emotional safety, engagement, and alignment with ASD intervention models (e.g., adjustable functions for sensory needs). For clinicians and special education teachers, it offers a structured protocol for integrating a novel tool into therapeutic practice, ensuring it complements established goals like joint attention or emotional regulation rather than disrupting them. The framework’s emphasis on qualitative data, triangulated through interviews, structured observations, and video analysis, ensures that the assessment captures the nuanced, individual responses of children with ASD, moving beyond mere technical functionality.

## Figures and Tables

**Figure 1 behavsci-15-01310-f001:**
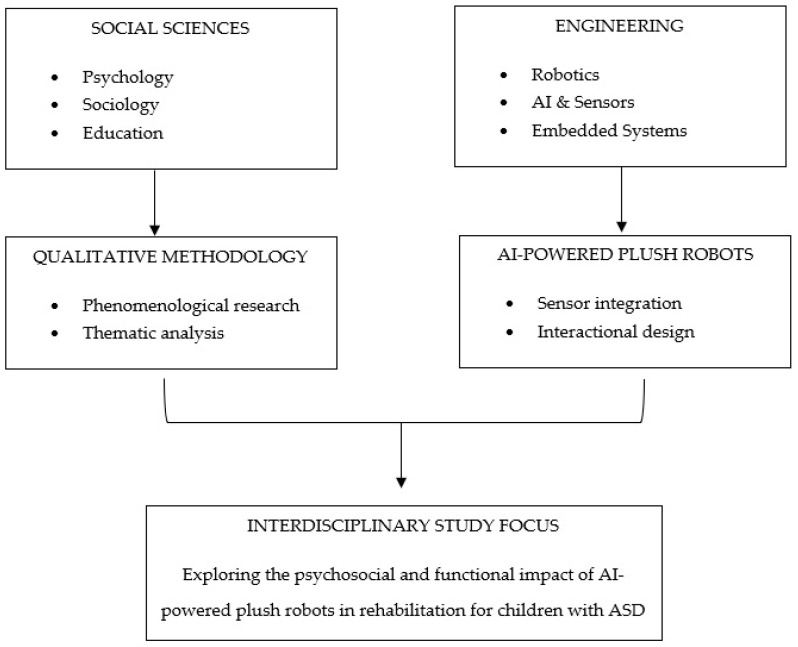
Interdisciplinary Research Framework on AI-Powered Plush Robots on Rehabilitation for Children with ASD.

**Figure 2 behavsci-15-01310-f002:**
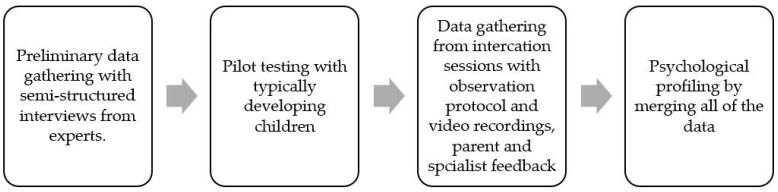
Sequential Framework for Pilot Testing and Psychometric Data Synthesis.

**Table 1 behavsci-15-01310-t001:** Integrated visual, functional, and psychological dimensions of assistive plush robots for ASD intervention.

Visual Properties	Functions	Psychosocial Aspects	Recommendations
Color	Sounds	Initial reaction and adjusting	Adjustable features
Size	Eye illumination	Similarity with pets	Function enhancement
Texture of the fabric and material	Area touch	Empathy	
Safe and suited for children with ASD	Reaction consistency	Training under adult supervision	
		Imitation	
		Language and speech development	
		Advantage for children with ASD	

## Data Availability

All collected video and audio data was stored securely on an encrypted server, accessible only to authorized members of the research team. Access to the data was password protected and required multi-factor authentication. In adherence to ethical guidelines and data protection regulations, all original video and audio recordings will be securely and irreversibly deleted following the completion of the data analysis and publication phases of this study.

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
