# Peer review of "Developing a Psychological Research Methodology for Evaluating AI-Powered Plush Robots in Education and Rehabilitation"

_behavsci, 2025, doi:10.3390/bs15101310_

Round 1
Reviewer 1 Report
Comments and Suggestions for Authors
It is always a pleasure to read a paper that makes a sincere and noble effort to support individuals with disabilities, particularly young people with Autism Spectrum Disorder (ASD), whose numbers are steadily increasing. I was especially pleased to see that the authors considered ethical aspects as an integral part of their research. The paper is well-structured and provides a clear overview of the entire project. I recommend acceptance for publication after minor revisions.
Comments and Suggestions:
Abstract Consistency: The abstract states that the study follows four phases. While phases 1, 2, and 4 are addressed to some extent, phase 3 is omitted. For the sake of clarity and consistency, I suggest briefly describing phase 3 in the abstract.
Related Work: The paper lacks a dedicated section on related work. Although some relevant studies are briefly mentioned in the introduction, which is an acceptable alternative, a few significant contributions appear to be missing. For example, the robot Kaspar has been effectively used in therapy for children with ASD. Additionally, during the COVID-19 pandemic, mobile and web applications were deployed to enhance accessibility. Including references to such initiatives would strengthen the background and contextual relevance of the study.
Subsection Announcements: Subsections 1.1 to 1.5 are not introduced in the main text. A few introductory sentences would improve readability and guide the reader through the structure of the paper.
Figures: The figures are clear but visually inconsistent with the rest of the manuscript. I recommend reformatting them to align with the MDPI style, particularly by using the appropriate font. Moreover, it is commendable that the figures are correctly numbered as Figure 1, Figure 2, etc., rather than by subsection.
Conclusion: The conclusion lacks rigor and does not clearly articulate how the proposed scientific methodology could be practically applied. Strengthening this section with concrete examples or recommendations would help ensure the work is not perceived as l’art pour l’art, but rather as a meaningful contribution with real-world applicability.
Author Response
Thank you very much for taking the time to review this manuscript. Please find the detailed responses below and the corresponding revisions highlighted changes in the re-submitted manuscript.
Comments 1: The abstract states that the study follows four phases. While phases 1, 2, and 4 are addressed to some extent, phase 3 is omitted. For the sake of clarity and consistency, I suggest briefly describing phase 3 in the abstract.
Response 1: Thank you for pointing this out. We agree that the original abstract lacked a description of Phase 3. Therefore, we have revised the abstract to include a concise summary of Phase 3, which involved structured interaction sessions with children diagnosed with ASD to evaluate the robot's rehabilitation potential. This change can be found in the updated abstract, specifically in the first paragraph, lines 22-24.
Comments2: The paper lacks a dedicated section on related work. Although some relevant studies are briefly mentioned in the introduction, which is an acceptable alternative, a few significant contributions appear to be missing. For example, the robot Kaspar has been effectively used in therapy for children with ASD. Additionally, during the COVID-19 pandemic, mobile and web applications were deployed to enhance accessibility. Including references to such initiatives would strengthen the background and contextual relevance of the study.
Respnse2: Agreed. We have addressed this by adding references to the Kaspar robot and mobile apps used during the COVID-19 pandemic in the Introduction, lines 48-52.
Comment3: Subsections 1.1 to 1.5 are not introduced in the main text. A few introductory sentences would improve readability and guide the reader through the structure of the paper.
Response3: We agree. We have supplemented the introduction preceding the subsections with a few sentences explaining their structure, as seen in lines 74-79 of the revised manuscript.
Comment4: The figures are clear but visually inconsistent with the rest of the manuscript. I recommend reformatting them to align with the MDPI style, particularly by using the appropriate font. Moreover, it is commendable that the figures are correctly numbered as Figure 1, Figure 2, etc., rather than by subsection.
Response4: We have reformatted all figures to align with the MDPI style guidelines, paying close attention to font type and size.
Comment5: The conclusion lacks rigor and does not clearly articulate how the proposed scientific methodology could be practically applied. Strengthening this section with concrete examples or recommendations would help ensure the work is not perceived as l’art pour l’art, but rather as a meaningful contribution with real-world applicability.
Response5: We have expanded the conclusion with concrete examples and recommendations, illustrating how the methodology can be practically applied in developing personalized therapy programs for children with ASD, lines 404-413.

Reviewer 2 Report
Comments and Suggestions for Authors
Resolve sample inconsistencies. Ensure Phase 2 sample size is consistently reported across abstract and methods.
-
-
The staged four-phase design is a strong contribution, but several methodological gaps limit transparency and reproducibility:
-
Sample sizes are inconsistently reported (Phase 2 TD children N=6–8 in the abstract vs. N=10–15 in Methods).
-
Clarify recruitment and inclusion criteria. Specify diagnostic verification source, recruitment sites, exclusion criteria, and whether purposive or maximum-variation sampling was used.
Operationalize instruments. Provide appendices with observation protocols, coding rubrics, and interview guides; define “sensor clusters” and “interaction clusters.”
-
- Instruments (“digital observation protocol,” “sensor clusters”) are referenced but not operationalized.
Report analytic reliability. Present inter-coder reliability and reflexivity statements; describe disagreement resolution procedures.
Detail procedures. Standardize session length, setting, specialist training content, and environmental controls; include fidelity checks.
-
- Session dosage, environmental controls, and specialist training are vague.
Introduce standardized measures. Supplement qualitative coding with validated scales (e.g., SUS, MARS, engagement coding rubrics, caregiver reports).
-
- No standardized usability or engagement measures are administered, despite being described in the theoretical framework.
Clarify triangulation. Explain how video, observation protocols, and interviews are integrated into final “psychological profiles.”
Balance claims. Emphasize preliminary, exploratory nature and avoid overgeneralization of therapeutic benefits.
Data governance. Expand on de-identification, storage, retention, and access to protect sensitive child video/audio data.
Reference accuracy. Consolidate duplicate citations and ensure formatting matches journal standards.
Author Response
Thank you very much for taking the time to review this manuscript. Please find the detailed responses below and the corresponding revisions highlighted changes in the re-submitted manuscript.
Comments1: Resolve sample inconsistencies. Ensure Phase 2 sample size is consistently reported across abstract and methods.
Response1: Implemented. The sample size for Phase 2 is now consistently reported as N=10-15 in both the Abstract and Methods sections.
Comments2: Clarify recruitment and inclusion criteria. Specify diagnostic verification source, recruitment sites, exclusion criteria, and whether purposive or maximum-variation sampling was used.
Response2: Thank you for pointing this out. We agree with this comment and have revised the manuscript accordingly. We have clarified the recruitment and inclusion criteria, specifying the diagnostic verification source, recruitment sites, exclusion criteria, and sampling method. These changes can be found in lines 203-215 of the revised manuscript.
Comments3: Operationalize instruments. Provide appendices with observation protocols, coding rubrics, and interview guides; define “sensor clusters” and “interaction clusters.”
Response3: Thank you for this helpful suggestion. We agree that operationalizing our instruments is critical for reproducibility. Therefore, we have: 1) Added appendices containing the observation protocols and interview guides; 2) Defined "sensor clusters" and "interaction clusters" within the manuscript text. These definitions can now be found in lines 185-198 of the revised manuscript, clarifying their operational use within our research.
Comments4: Instruments (“digital observation protocol,” “sensor clusters”) are referenced but not operationalized.
Report analytic reliability. Present inter-coder reliability and reflexivity statements; describe disagreement resolution procedures.
Response4: Thank you for raising this important point. We agree that detailing our analytical process and ensuring reliability is essential. Therefore, we included a description of our analytic process, including procedures to address inter-coder reliability and reflexivity, and detailing disagreement resolution methods. The procedures regarding data reliability and for our instrument "digital observation protocol" and "sensor clusters" is now described in Subsection 2.1. lines 167-179 of the revised manuscript.
Comments5: Detail procedures. Standardize session length, setting, specialist training content, and environmental controls; include fidelity checks.
Response5: Thank you for highlighting the need for more detailed procedural information. We agree that standardization is important for rigor. Therefore, we have added detail regarding session length, setting, specialist training content, environmental controls, and fidelity checks. This information can be found in lines 215-231 of the revised manuscript.
Comments6: Session dosage, environmental controls, and specialist training are vague.
Introduce standardized measures. Supplement qualitative coding with validated scales (e.g., SUS, MARS, engagement coding rubrics, caregiver reports).
Response6: We appreciate this feedback and have addressed this point in the revised manuscript.
We appreciate this insightful suggestion about validated scales. We agree that the inclusion of standardized measures will strengthen the rigor of our evaluation. Therefore, we have incorporated the System Usability Scale (SUS) into our data collection process. Details regarding the administration of the SUS can be found in Section 2.3 lines 219-231.
Comments7: No standardized usability or engagement measures are administered, despite being described in the theoretical framework.
Comments7: We acknowledge this concern and have addressed it accordingly. Please see Comment 6 for a detailed description of the implemented improvements, which are included in subsection 2.3
Comments8: Clarify triangulation. Explain how video, observation protocols, and interviews are integrated into final “psychological profiles.”
Response8: Thank you for raising this point about clarifying our triangulation process. We have addressed this in detail, please see Response 4 for an explanation of how video data, observation protocols, and interviews were integrated into the final "psychological profiles." These changes can be viewed in subsection 2.1. lines 167-179.
Comments9: Balance claims. Emphasize preliminary, exploratory nature and avoid overgeneralization of therapeutic benefits.
Response9: We acknowledge this comment and have carefully revised the discussion section to emphasize the preliminary and exploratory nature of our findings, avoiding overgeneralizations regarding therapeutic benefits
Comments10: Data governance. Expand on de-identification, storage, retention, and access to protect sensitive child video/audio data.
Comments10: We appreciate this important feedback. We have expanded the information in the "Data Accessibility Statement" regarding our data management procedures and permanent video deletion in lines 432-436.
Comments11: Reference accuracy. Consolidate duplicate citations and ensure formatting matches journal standards.
Response11: We appreciate you bringing this to our attention. We have carefully reviewed all references and corrected any inconsistencies in formatting to align with the journal's style guidelines. While some sources may appear to be duplicates due to shared authors, these are in fact distinct publications with differing publication dates, and as such they are not redundant.
Reviewer 3 Report
Comments and Suggestions for Authors
A very well-written, impactful article. The aim to produce an appropriate interdisciplinary methodology to evaluate usability, impact, and therapeutic efficacy of AI Soft Plush robots for ASD children was relevant, original and important. The research was clearly described and rigorous.
The only recommendation I would make is for the authors to highlight the importance of their research. Your conclusion is very short and would benefit from from a short discussion on the benefits of these soft plush AI toys for ASD children. Then link it to the importance of evaluating these toys.
Finally, could we have some pictures of the AI plush toys?
Author Response
Thank you very much for taking the time to review this manuscript. Please find the detailed responses below and the corresponding revisions highlighted changes in the re-submitted manuscript.
Comments1: The only recommendation I would make is for the authors to highlight the importance of their research. Your conclusion is very short and would benefit from from a short discussion on the benefits of these soft plush AI toys for ASD children. Then link it to the importance of evaluating these toys.
Response1: Agreed. We have added more emphasis on this in the conclusion of the article lines 386-513.
Comments2: Finally, could we have some pictures of the AI plush toys?
Response 2: We thank the reviewer for their interest in our AI-powered plush robot prototypes. Visual examples of the four plush robot designs are now included in Appendix 2 (observation protocols), and have also been published in a related article that details the design process and expert evaluation: Bethere, D., Tīģere, I., Hofmane, A., Šteinberga, A., Gavriļenko, U., Meļķe, S., Okss, A., Kataševs, A., & Vališevskis, A. (2024). AI-Powered Plush Robots for Children with ASD in Education, Rehabilitation: Expert Evaluation. IAFOR Journal of Education, 13(2). https://doi.org/10.22492/ije.13.2.03

Round 2
Reviewer 2 Report
Comments and Suggestions for Authors
Only interview protocols and observation forms are included in the appendicies. The coding rubrics are essential for replication and should be added.
The code reconciliation process is described, but quantitative reliability measures (e.g., Cohen's kappa) should be reported to demonstrate the actual reliability achieved.
The integration of video data, observation protocols, and interviews into "psychological profiles" needs more explicit description. How exactly were these data sources combined?
The study primarily validates methodology rather than demonstrating robot effectiveness. Consider tempering claims about therapeutic potential until more substantial empirical evidence is available.
Author Response
Comments1: Only interview protocols and observation forms are included in the appendicies. The coding rubrics are essential for replication and should be added.
Response1: We thank the reviewer for this valuable comment. In our study, we employed reflexive thematic analysis, where categories are generated inductively from the data rather than derived from a predefined coding rubric. As such, no fixed coding scheme was available to include in the appendices. The three experts independently developed initial themes and subthemes, which were subsequently refined through discussion and consensus-building. Because coding categories were not established a priori, intercoder reliability coefficients such as Cohen’s kappa are not applicable in this context. Instead, rigor and trustworthiness were ensured through triangulation of researchers, iterative analytic discussions, and transparency of the analytic process. We made the corrections in paragraph 2.1., page no 4: in line 175 about this being reflexive thematic analysis and in lines 164-167 emphasized that the categories in Appendix 1 are not a set of predefined categories.
Comments2: The code reconciliation process is described, but quantitative reliability measures (e.g., Cohen's kappa) should be reported to demonstrate the actual reliability achieved.
Response2: Thank you very much for pointing this out, as we understand according to Braun and Clarke’s guidelines such inter-rater reliability calculations as Cohen’s kappa or Krippendorff’s alpha is a great reliability coefficient for preexisting categories, which can be the case in content analysis. However in the guidelines for reflexive thematic analysis, intercoder reliability coefficients are not applicable, because coding is conceptualized as a subjective, interpretative process rather than a standardized procedure. Instead of quantifying agreement, rigor was ensured through researcher triangulation, iterative discussions to refine coding and themes, reflexive engagement with the data, and transparency in documenting analytic decisions. We thank the reviewer for this valuable remark and thus have emphazised this discrepancy in in paragraph 2.1., page no 4: in lines 190-193.
Comments3: The integration of video data, observation protocols, and interviews into "psychological profiles" needs more explicit description. How exactly were these data sources combined?
Response3: Thank you for this helpful suggestion, we have detailed the description in section 2.4. lines 244 – 269.
Comments4: The study primarily validates methodology rather than demonstrating robot effectiveness. Consider tempering claims about therapeutic potential until more substantial empirical evidence is available.
Response4: Thank you for the valuable feedback. We fully agree and have implemented your suggestions by tempering claims about therapeutic potential and refocusing on methodological contributions. The key revisions are highlighted in red in both the Abstract and Discussion sections.
